# Can Foundation Models Watch, Talk and Guide You
# Step by Step to Make a Cake?

**Yuwei Bao[†], Keunwoo Peter Yu[†], Yichi Zhang[†], Shane Storks[†], Itamar Bar-Yossef[†],**
**Alexander De La Iglesia[†], Megan Su[†], Xiao Lin Zheng[§][*], Joyce Chai[†]**
[†]University of Michigan, [§]Syracuse University

## Abstract

Despite tremendous advances in AI, it remains a significant challenge to develop interactive task guidance systems that can offer situated, personalized guidance and assist humans in various tasks. These systems need to have a sophisticated understanding of the user as well as the environment, and make timely accurate decisions on when and what to say. To address this issue, we created a new multimodal benchmark dataset, **Watch, Talk and Guide (WTaG)** based on natural interaction between a human user and a human instructor. We further proposed two tasks: *User and Environment Understanding*, and *Instructor Decision Making*. We leveraged several foundation models to study to what extent these models can be quickly adapted to perceptually enabled task guidance. Our quantitative, qualitative, and human evaluation results show that these models can demonstrate fair performances in some cases with no task-specific training, but a fast and reliable adaptation remains a significant challenge. Our benchmark and baselines will provide a stepping stone for future work on situated task guidance.

## 1 Introduction

You have probably watched a lot of YouTube videos on how to bake a cheesecake, or how to change the car windshield wipers, but something always goes wrong and you just wish there is an expert right there to guide you through. Can we design an artificial intelligent system to watch, talk and guide humans step by step to complete a given task?

Task guidance for human users is a challenging problem, as it requires an interactive system to have a sophisticated understanding of what the user is doing, under the environment setup, and providing appropriate timely guidance. What is more challenging is if we could design a model

that can be easily generalized to any arbitrary task without prior exposure, given a task manual or one demonstration. This requires the model to have a robust knowledge base and in-context learning abilities to easily pick up a new task to guide the human through.

Traditional approaches to develop AI agent for interactive task guidance like this require a large amount of task-specific training or rules to recognize object states (Gao et al., 2016), mistakes in actions (Du et al., 2023), and to interact with humans (Wu et al., 2021), thus limiting their ability to generalize. However, the recent rise of foundation models trained on a large amount of multimodal data from the web (Brown et al., 2020; Radford et al., 2021b; Alayrac et al., 2022; Li et al., 2022, 2023) creates new opportunities for developing robust open-domain AI agents for this problem. These works have undergone a paradigm shift and begun to explore the zero- and few-shot adaptation of these models to various embodied AI problems (Khandelwal et al., 2022; Huang et al., 2022; Ahn et al., 2022; Kapelyukh et al., 2023).

In this work, we extend this paradigm shift by examining the application of recent state-of-the-art foundation models to situated interactive task guidance. We created **Watch, Talk and Guide (WTaG)**, a new multimodal benchmark dataset which includes richly annotated human-human dialog interactions, dialog intentions, steps, and mistakes to support this effort. We define two tasks: (1) *User and Environment Understanding*, and (2) *Instructor Decision Making* to quantitatively and qualitatively evaluate models' task guidance performance. With the inherent complexity of the problem itself, this dataset can help researchers understand the various nuances of the problem in the most realistic human-human interaction setting. We used a large language model (LLM) as the backbone for guidance generation, and explored three different mutimodal methods to extract vi-

---

[*]Work done during a summer internship at the University of Michigan.

sual and dialog context. Our empirical results have shown promising results with foundation models, and highlighted some challenges and exciting areas of improvement for future research. The dataset and code are available at `https://github.com/sled-group/Watch-Talk-and-Guide`.

## 2 Related Work

### 2.1 Task Guidance Systems

Traditional task guidance systems (Ockerman and Pritchett, 1998, 2000) focused on providing the user with pre-loaded task-specific information without tracking the current state of the environment, or being able to generalize to new tasks (Leelasawassuk et al., 2017; Reyes et al., 2020; Lu and Mayol-Cuevas, 2019; Wang et al., 2016). The complexity of the problem comes from various aspects, such as environment understanding, object and action recognition, user's preference and mental state detection, real-time inference, etc (Manuvinakurike et al., 2018; Kim et al., 2022). In this work, we collected a multimodal dataset with real human-human task guidance interactions to better study the depth and breadth of the problem. We established a strong zero-shot baseline without prior exposure of the given tasks and develop a task guidance system with the help of the latest AR advancement to incorporates both users' perception and dialog.

### 2.2 Language and Multimodal Foundation Models

Large language foundation models (LLMs) such as ChatGPT,[1] GPT-4 (OpenAI, 2023a), and Bard[2] have demonstrated a wide range of language generation and reasoning capabilities. These models are not only equipped with huge knowledge bases through training on web-scale datasets, but also the in-context learning ability that allows them to learn new tasks from a few examples without any parameter updates (Brown et al., 2020).

Meanwhile, multimodal foundation models such as GPT-4v (OpenAI, 2023b), CLIP (Radford et al., 2021a) are also on the rise to incorporate large scale vision (Betker et al., 2023; Peng et al., 2023; Zhang et al., 2022; Li et al., 2022, 2023), audio (OpenAI, 2022), embodied (Brohan et al., 2023) and other input modalities (Zellers et al., 2021, 2022; Radford et al., 2021b; Li et al., 2022, 2023; Alayrac et al.,

[1]`https://openai.com/blog/chatgpt`
[2]`https://ai.google/static/documents/google-about-bard.pdf`

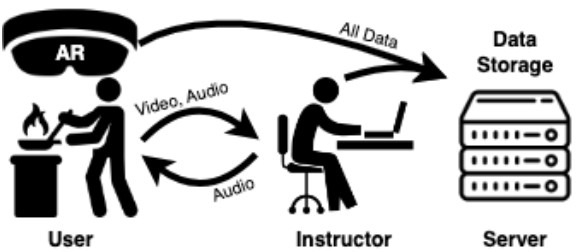

Figure 1: Data collection setup for WTaG. Human users follow simplified recipes while communicating with the human instructor through an augmented reality device. Human instructors watch the egocentric view from the user and offer guidance based on detailed recipes. All data streams are recorded for further analysis.

2022; Moon et al., 2020) that can reason and generate one modality type given another. While some proprietary models can be difficult or expensive to access (e.g. GPT-4v), other open source multimodal foundation models have been adapted to problems related to task guidance systems, e.g., action success detection (Du et al., 2023). In this work, we leverage the large-pretrained world knowledge embedded in these foundation models, and LLMs' in context learning ability, to build a generalizable situated task guidance system.

## 3 A Dataset for Situated Task Guidance

In this work, we introduce **Watch, Talk, and Guide (WTaG)**, a new dataset for situated task guidance. WTaG includes nearly 10 hours of egocentric videos of human users performing cooking tasks while guided by human instructors through live, natural interaction. Synchronized videos and audio transcripts in WTaG present a variety of challenging phenomena, including perceptual understanding, communications, natural mistakes, and much more. We hope this dataset can serve as the starting point to dive into this complex problem.

Egocentric video datasets (Table 1) have garnered much attention in the past decade thanks to their potential application in interesting research areas such as embodied AI and task guidance systems. Most of the large egocentric video datasets contain unscripted activities (Damen et al., 2018, 2020; Lee et al., 2012; Su and Grauman, 2016; Pirsiavash and Ramanan, 2012; Fathi et al., 2012; Grauman et al., 2022), while others have collected (semi-)scripted activities where camera wearers are asked to follow certain instructions (Sigurdsson et al., 2018; Li et al., 2018; Sener et al., 2022). Meanwhile, related works in embodied AI have collected similar

| Dataset | Statistics | | Characteristics | | | Annotations | | |
|---|---|---|---|---|---|---|---|---|
| | Hours | Task Sessions | Natural | Interactive | Mistakes | Action Descriptions | Dialog Intents | Mistake Types |
| ADL (Pirsiavash and Ramanan, 2012) | 10 | 20 | ✓ | | | ✓ | | |
| Charades-Ego (Sigurdsson et al., 2018) | 69 | 68.5K | ✓ | | | ✓ | | |
| EGTEA Gaze+ (Li et al., 2018) | 28 | 86 | ✓ | | | ✓ | | |
| Epic-Kitchens (Damen et al., 2018) | 55 | 432 | ✓ | | | ✓ | | |
| Ego4D (Grauman et al., 2022) | 3.7K | 931 | ✓ | | | ✓ | | |
| Assembly101 (Sener et al., 2022) | 513 | 362 | ✓ | | ✓ | ✓ | | ✓ |
| ALFRED (Shridhar et al., 2020) | – | 8.1K | | | | ✓ | | |
| MindCraft (Bara et al., 2021) | 12 | 100 | | ✓ | | ✓ | | |
| TEACh (Padmakumar et al., 2022) | – | 3.2K | | ✓ | ✓ | ✓ | | |
| **WTaG (Ours)** | 10 | 56 | ✓ | ✓ | ✓ | ✓ | ✓ | ✓ |

Table 1: Comparison of WTaG to past egocentric task-oriented video datasets. We compare datasets in terms of data statistics, dataset characteristics (whether videos are natural, involve dialog interaction, or have annotated mistakes), and types of annotation (action type or narration, dialog act categorization, or mistake details) available.

collaborative task completion datasets generated through virtual environment simulators (Shridhar et al., 2020; Bara et al., 2021; Padmakumar et al., 2022). They also target task-oriented dialog and mistakes but are not as natural nor can be generalized to the open world as can our work.

While smaller than some existing datasets (Table 1), WTaG emphasizes the interactions between the user and the instructor, prioritizes depth over breadth with this uniquely rich dataset for situated task guidance, and is (to our knowledge) the first of its kind with natural, human-to-human interactive videos annotated with conversations texts, recipe steps, dialog intents, and mistakes.

## 3.1 Data Collection

To collect data, two human subjects, an *instructor* and a *user*, are paired up. The user is tasked with completing 1 of 3 cooking recipes[3] while communicating with an instructor. To encourage natural mistakes and interaction between the instructor and user to occur, the instructor has access to a complete, detailed ground truth recipe, while the user only has a simplified version of it, with high-level directions and minimal details (Figure 9).

As shown in Figure 1, the instructor is separated from the user during task completion, watching and communicating with them through an egocentric camera view interface and external microphone connected to an augmented reality (AR) headset[4]

[3]Recipes include peanut butter and jelly pinwheels, pour-over coffee, and a microwaved mug cake.

[4]We use Microsoft HoloLens 2 (https://www.microsoft.com/en-us/hololens/), but such interaction can be enabled by any wearable audiovisual device.

worn by the user. We use Microsoft Azure automatic speech recognition (ASR)[5] to convert audio recordings from videos into conversations transcripts, and manually corrected them as needed. All data from both sides are synchronized and sent to a server for storage and future processing. Although not used in our experiments, we also collect 12 additional types of synchronized data using Microsoft Psi on top of the egocentric RGB video, user and instructor audios for each recording. More details can be found at https://github.com/microsoft/psi.

A total of 56 recordings were collected from 17 user subjects and 3 instructor subjects. All human subjects were over the age of 18, English-speaking college students with different genders and are from different cultural background with normal or corrected-to-normal vision, recruited through messaging platforms. Members of the study team served as the human instructors.

Together, 4,233 English dialog utterances were collected, evenly distributed over the 3 recipes (19 pinwheels, 18 coffees, 19 cakes). The length of the videos range from 5 to 18 minutes, with the median of 10 minutes. As shown in Figure 2, each video contains a median of 31 instructor utterances and 35 user utterances. The instructors talk a bit longer than the users in each utterance, with a median of 6 words per utterance versus 3 words for users. Instructors also speak faster than the users with a median speed of 398 ms/word versus 522 ms/word.

[5]https://azure.microsoft.com/en-us/products/cognitive-services/speech-to-text/

![WTaG dataset statistics histograms]

Figure 2: WTaG dataset statistics histogram for videos and dialog interactions.

## 3.2 Data Annotation

To have an in depth understanding on how human instructors navigate this complex task and nuanced situations, and an easier way to evaluate models' performance on WTaG, we provide rich annotations on recordings. First, we annotate the time span of each recipe step the user performs in each video, facilitating user state detection. Secondly, we manually go through all the ASR results, correct the transcripts and voice activity time span as needed, and filter out any potential harmful speech. Lastly, we categorize user and instructor utterances into a set of dialog intentions, together with mistake types if any. Details are as follows:

**User utterances** are categorized into 6 intents: *Question*, *Answer*, *Confirmation*, *Self Description*, *Hesitation*, and *Other*.

**User mistakes** are categorized into 3 classes: *Wrong Action*, *Wrong Object*, and *Wrong State* (including measurement and intensity).

**Instructor utterances** are categorized into the following 5 coarse-grained intents: *Instruction*, *Question*, *Answer*, *Confirmation*, and *Other*.

If the instructor decided to issue an "instruction", the **instructions** are further classified into 4 types based on what they inform users about: *Mistake Correction*, *Current Step*, *Next Step*, and *Details*.

## 4 Task Definitions

For benchmark evaluation, we extract query points from the WTaG dataset whenever one of the following conditions is met:

(a) GT user said something

(b) GT instructor said something

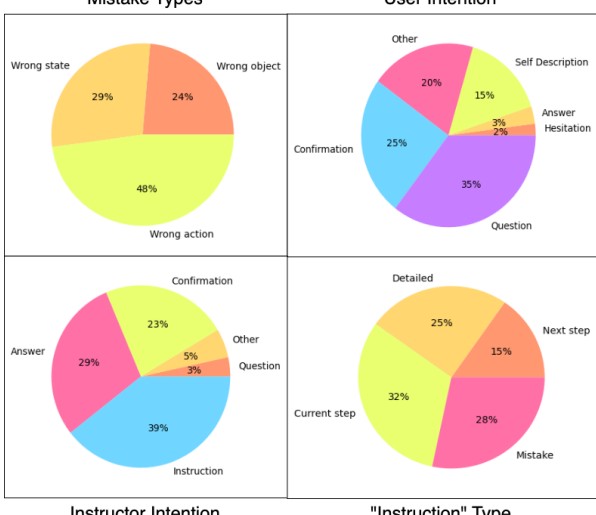

Figure 3: User and Instructor Dialog Intention Type Distributions and Mistake Type Distributions of WTaG

(c) GT no one said anything for 10 seconds

Each query point provides systems with the latest frame image, dialog history, the task recipe, and current elapsed time into the task. This creates a variety of situations, where the instructor may or may not need to intervene and provide guidance.

More specifically, for each query time point $t$, given the user's egocentric video frame, and the chat history, we formulate the following two tasks for the models to predict.

**User and Environment Understanding**

1. User intent prediction: Dialog intent of user's last utterance, if any (options).

2. Step detection: Current step (options).

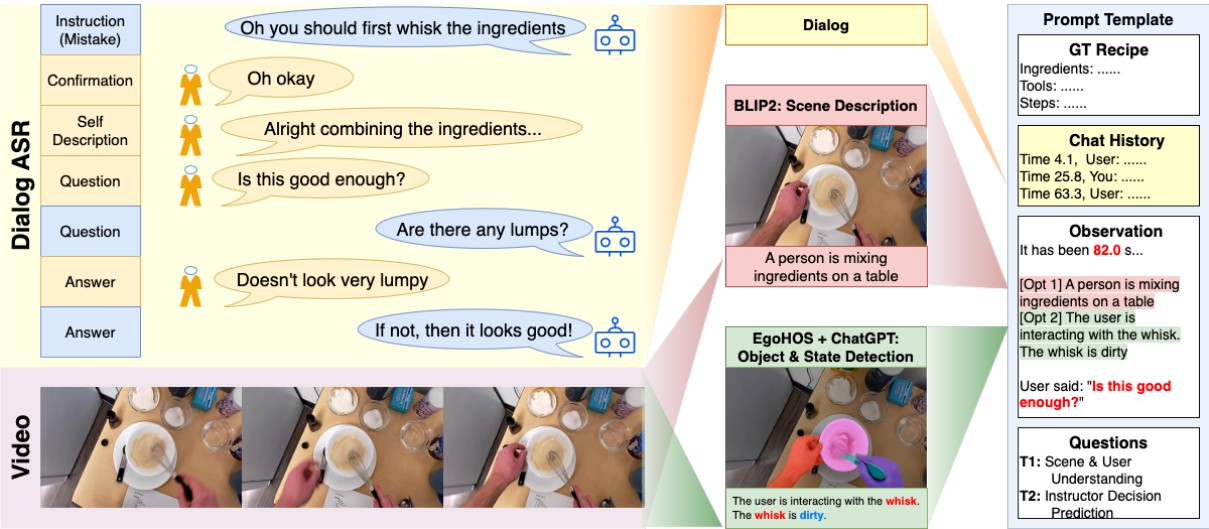

Figure 4: Interactive Task Guidance Pipeline: Synchronized video and dialog transcripts are inputs to the system. We annotated each utterance to reflect dialog intent. The dialog history is inserted into the template, and the latest utterance is part of the observation. To process the videos, each queried frame either goes through BLIP2 for a scene description, or goes through EgoHOS for object and state detection. Zero, or one of the two video extraction output is inserted into the prompt template. The prompts are sent to ChatGPT for instruction predictions.

3. Mistake Existence and Mistake Type: Did the user make a mistake at time $t$ (yes/no). If so, what type of mistake (options).

**Instructor Decision Making**

1. When to Talk: Should the instructor talk at time $t$ (yes/no).

2. Instructor Intent: If yes to 1, instructor's dialog intention (options).

3. Instruction Type: If yes to 1 and intent in 2 is "Instruction", what type (options).

4. Guidance generation: If yes to 1, what to say in natural language.

## 5 Methods

Given the WTaG dataset and the two tasks defined above, we explore the application of pre-trained large language and vision foundation models on this problem *without task-specific training*.

For each recording in WTaG, we feed the video frame by frame to our system, together with the synchronized dialog transcripts, and only query ChatGPT at the three query conditions (Section 4). The visual frames go through optional multimodal information extraction process detailed below, to translate the relevant visual context into natural language. The dialog transcripts are extracted at the first frame each utterance appears, and offer conversational context for model predictions. We

designed a prompt template (Figure 4) that includes the ground truth recipe template, user-instructor chat history up to time point $t$, observations of the user and environment, as well as a list of questions listed in Section 4. We then send the prompts to ChatGPT[6] to answer questions related to the proposed tasks in each query time point. To enrich the prompts and offer more context, we explored the following three methods that translate multimodal precepts from the egocentric video into language:

**Language Only (Lan):** As a baseline for the LLM, we extracted the ground truth user and instructor dialog up until time $t$. All the past utterances are added to the prompt as part of the interaction history, and only the most recent user utterance is added to the observation to avoid model cheating. To enable temporal reasoning, observations include how long the user has been following the recipe so far. This method does not offer any visually-dependent information to the LLM backbone, and challenges the model to infer the context purely based on the conversation.

**Scene Description (Sce):** In this method, we generate a free-text scene description by applying BLIP-2 (Li et al., 2023) to the latest frame image, and insert it into the prompt as part of the observations. Depending on the prompts to BLIP-2, we ask generic questions such as "What is the user do-

---

[6]https://openai.com/blog/chatgpt; used Azure endpoint for GPT3.5-turbo-0301, which is trained on data from up to September 2021.

ing" or "This is a picture of" to get the descriptions. While this approach is flexible and open-domain, there is no control of how much situation-specific the descriptions would be. Together with the time lapsed and the dialog history, this method offers more scene level visual context to the LLM.

**Object and State Detection (Obj)**: In this method, we extract more fine-grained scene information by detecting important objects in the frame image and their corresponding states, and inserting them into the prompt observations. At each query time $t$, we use EgoHOS (Zhang et al., 2022) to segment user's hands and the objects that they are interacting with. We then extract the object segments and predict their most likely object names and their corresponding states using CLIP (Radford et al., 2021b). To balance the performance and generalizability of CLIP predictions, we first extracted a list of most likely objects and their potential states from the recipe through a separate prompt to the LLM backbone. The resulting list of objects and states go through a minor manual cleanup before being used by CLIP. This narrows down the scope of search for CLIP, and offers better targeted vision to language prediction, but is also easily generalizable to open-world object and state detections.

All three methods were queried at the same time points as extracted in Section 4 for fair comparison. We reserved 6 recordings (2 of each recipe) for hyperparameter and prompt tuning, and the rest for evaluation. Each method was repeated 3 times. ChatGPT APIs configurations are: Max tokens=100, temperature=0, stop words= [\n \n, —, """, '"]. All experiments conducted on a single GPU NVIDIA RTX A6000 and a Intel(R) Core(TM) i9-10900X CPU @ 3.70GHz.

# 6 Experimental Results

In this section, we evaluate the three methods above on the following two tasks: *User and Environment Understanding*, and *Instructor Decision Making*. Micro F1 scores are reported for each classification task. We further break down the performance, and evaluate how accurately each vision extraction module can translate the scene into language, and how helpful or annoying the model's generated sentences are under a human evaluation.

Figure 5: Interactive Task Guidance Micro F1 Scores: For user and environment understand tasks, the models demonstrated well above random chance performance in user intention prediction and step detection, but struggle with mistake recognition. For Instructor decision prediction tasks, the models showed above random chance performance in predicting instructor's intention and instruction types, but issued higher communication frequencies than human instructors. Across the three methods, Language Only (Lan) showed comparable performance even without any visual context.

## 6.1 User and Environment Understanding

As an interactive task guidance agent, it is important for the model to have a comprehensive understanding of the task's physical environment, as well as users' mental and physical states, through their conversations as well as actions.

The overall user utterance intention prediction, step detection, mistake recognition and type prediction performances can be found in Figure 5a,5b. It was observed that all three methods using zero-shot foundation models have demonstrated decent performance significantly above the random guessing (grey dash line) on user intention predictions and step recognition, but struggled with mistake detection. This is most likely due to the limited visual context the models can offer to accurately detect mistakes (More in Section 6.3). Out of the ones that the model did correctly predict that a mistake has happened, it displayed chance level of performance.

Among the three methods, it is observed that with just the dialog context alone, the model was

able to achieve comparable performance as the other two. This shows that the conversations between the human user and instructor disclose a lot of information about the user and the environment even without visual perception inputs. The Object and State Detection (Obj) method outperforms the Language Only method by a small but significant margin on the user intention and step prediction tasks. Compared to Obj, the Scene Description (Sce) showed worse and more inconsistent performance across the tasks. It is likely that the visual context this method extracts is too generic or hallucinating, which on the contrary confuses the LLM predictions. We further evaluated how well these visual extraction modules perform in Section 6.3.

## 6.2 Instructor Decision Making

Given the context of what the user is doing and what the task environment is at each query point during task execution, the instructor needs to provide situated guidance on when to talk, and what to say in terms of intents and free-formed guidance.

We evaluate if the models can correctly predict "when to talk" at each query point, based on if there is a ground truth instructor utterance in the next $x$ seconds, where $x = \text{median}(\text{Inst}_{speed}) \times \text{median}(\text{Inst}_{wd/uttr})$. The model could decide if it needs to talk triggered by user's utterance, previous ground truth instructor's comment, and visual context when no one talks. To evaluate models' decision making performance on "what to talk" (dialog intention and free-formed guidance), we collect models' predictions at each query condition (b), i.e. whenever the GT instructor talked. Note, the GT instructor utterances were not part of the prompt to prevent information leakage.

The overall instructor decision prediction can be found in Figure 5c. It was observed that all three methods predicted when to talk with around chance performance. Going through the examples (Figure 10, 11), we have found that the ChatGPT has a stronger tendency to offer more frequent guidance when the ground truth humans don't. All three methods have a significant above random chance aligning with the ground truth instructor and instruction intentions. Similar to user and environment understanding performances, the Language Only method demonstrated strong performance, especially in deciding the instructor's intention, whereas the Sce method fell short across all three subtasks.

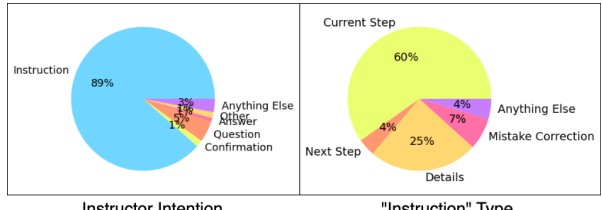

Figure 6: Dialog Intention Prediction Distribution: Chat-GPT has a strong tendency to issue instructions, and especially instructions about the current step to the users. Compared to human instructors (Figure 3), the guidance is less situated, personalized, or natural.

Human language is rich and diverse and there are usually more than one acceptable way of guiding the users. Therefore, we further looked into the distributions of the model intention predictions. Comparing Figure 6 with the human intention distribution in Figure 3, the LLM tends to issue a lot more instructions to users than humans do; human instructors offer more diverse responses, including more answers and confirmation. Among all the instructions, the models tend to describe the "Current Step" whereas human instructors describe more evenly distributed instruction types. With limited user modeling and visual understanding, it is understandably harder for LLMs to offer situation-specific responses, and therefore resort to more generic instructions about the current step.

Lastly, we conducted a human evaluation on models' generated language guidance.

Situated interactive task guidance is a personalized challenge. The guidance frequency and content vary a lot from user to user, according to their familiarity with the task itself, their chattiness, mental states, etc. We broke down the performance based on the number of dialog utterances that occurred in recordings into three categories: **short**, **mid**, and **long**. All test recordings were evenly divided accordingly. In this section, we selected 6 test recordings (2 per recipe), and asked 3 human evaluators with different genders and cultural backgrounds to rate the following for each output:

1. How helpful do you find this instructor utterance is? Rate 1/2/3, 3 = Very Helpful
2. How annoying do you think it is? Rate 1/2/3, 3 = Not Annoying

A total of 936 time points were evaluated, and each time point was rated by two annotators.

The aggregated results for the three methods are shown in Figure 7. Overall, the average quality

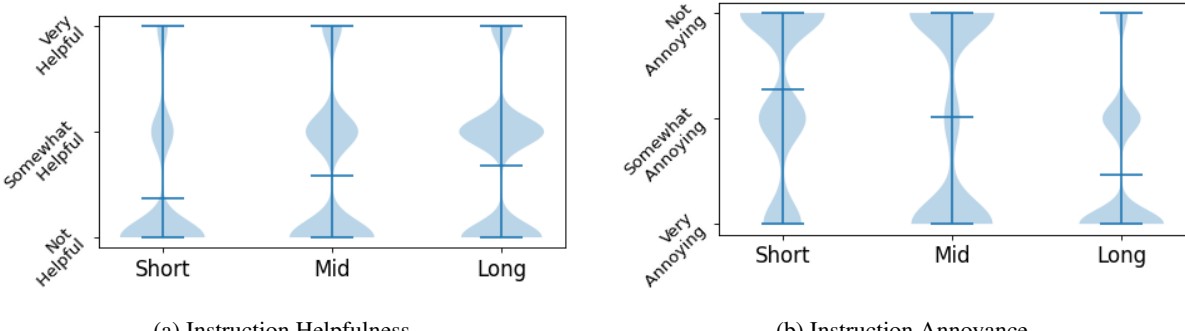

| (a) Instruction Helpfulness | (b) Instruction Annoyance |

Figure 7: Human Evaluation on Model Generated Guidance: Overall on average, most guidance are not considered as very helpful, and somewhat to very annoying by human evaluators. Although not significant, more guidance in shorter videos are found to be less helpful, and more in longer videos are labeled as very annoying. In general, task guidance can be a very personalized experience.

of the generated instructions was not considered as very helpful by the human evaluators. They are somewhat to very annoying. This shows that situated natural language task guidance still has a long way to go to be applicable even though they've shown above random chance level performances across most of the tasks evaluated. Among the three groups, although not significant, more guidance in the short videos were considered as "Not Helpful" by humans, whereas more guidance in the long videos were thought as "Very Annoying".

We also measured the inner annotator agreement on these results, and calculated the Cohen's $\kappa$ (Cohen, 1960) value for helpfulness (0.14) and annoyance (0.02) ratings. This shows that the task guidance can be a very personal experience and that different users have different preferences and tolerance on the guidance. While the methods we experimented with here did not inject user preferences into the prompts, and the LLM was unable to detect nuanced user mental preferences through the dialog, future work may begin to study how foundation models can be used for these challenges.

### 6.3 Perceptual Input Extraction

In order for the LLM backbone model to have an accurate assessment of the user and the environment, a high quality visual perception extraction module is a necessity. In this experiment, we did an ablation study on how truthful the Scene Description (Sce) and the Object and State Detection (Obj) modules are while translating the perceptual inputs to language. We conducted a small scale human evaluation on 6 recordings (2 of each recipe) and evaluated the truthfulness of the vision outputs. The truthfulness is defined as whether the vision output of each scene contains information

that are not present or completely irrelevant to the scene. For example, "a person is putting ketchup on a plate" for the pinwheel recipe in the WTaG dataset is not truthful, as this action is not part of any recording. This metric is more factual based than subjective, and all results below were evaluated by one person.

For the Scene Description evaluation (Figure 8a), we tried out 3 different prompts to the BLIP-2 model: no prompt (only the image), "Question: What is the user doing? Answer:", and "This is a picture of". There wasn't a significant difference in truthfulness among the three prompts, and we went with no prompt for all the experiments. However, overall, it was observed that the truthfulness of BLIP-2 is below 30%, which means most of the scene descriptions are actually hallucinating. This could cause huge confusions to the LLMs about exactly what is happening in the scene.

For the Object and State Detection evaluation (Figure 8b), since EgoHOS outputs unstable predictions from frame to frame, we conducted an experiment on detected object smoothing strategies. For a sliding window of 10 frames, we tried out if the same object needs to occur from 1 to 5 times in order to be included in the LLM prompts. For each prompt query, we evaluate 1) if any object and state were detected at all from the module, and 2) if the detection results were truthful. According to Figure 8b, it was observed that as the required smooth occurrence goes higher, the same object needs to be detected more frequently to be included in the output; while at the same time, fewer objects were detected throughout the recipe overall. We picked smooth rate of 2 for all the experiments. As a result, the visual detection is about 70% accurate with the this method. It is much higher than the BLIP-2

| Prompt | Truthfulness |
|---|---|
| None | 26.2±11.9 |
| "Question: What is the user doing? Answer:" | 25.4±13.2 |
| "This is a picture of" | 25.3±12.1 |

(a) BLIP2 Scene Description Evaluation

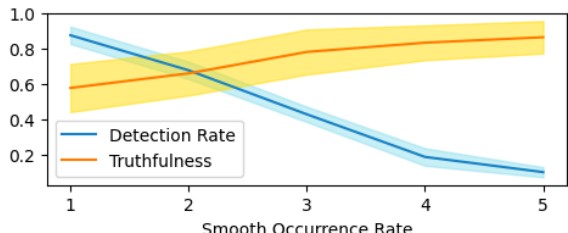

(b) EgoHOS Object and State Detection Evaluation

Figure 8: Vision to Language Translation Performance. (a) BLIP2 scene descriptions are reported to be only about 25% truthful. (b) A smooth occurrence rate of 2 is chosen for the experiments and offers about 70% truthfulness of the extracted visual context.

Scene Description method, but still noisy enough to confuse the language model.

## 7 Conclusion and Discussions

In this work, we explored how to leverage foundation models to guide human users step by step to complete a task in a zero-shot setting given an arbitrary task recipe. We created a new benchmark dataset, Watch, Talk and Guide (WTaG), with natural human user and instructor interactions, and mistake guidance. Two tasks were proposed to evaluate model's ability on user and environment understanding, as well as instructor decision making. We used a large language model as the backbone for guidance generation, and compared three configurations of inputs incorporating language and visual context through dialog history, scene description, and object and state detection with multimodal foundation models. We conducted quantitative, and human evaluations of the three methods on our dataset, and discovered several challenges for future work.

First of all, vision to language translation is challenging. Having an accurate and relevant description of what the user is doing and the environment setup is non-trivial. Some of the methods we've experimented with often describe untruthful scenes or objects, or offer true but too generic or irrelevant information, e.g., "the user is preparing food." For future work, it would be helpful if these vision to language models could leverage recipe information, and user's attention (e.g., eye gaze), and other sensory inputs (e.g., sound) to achieve richer and more relevant user and environment descriptions.

Second, the overall performance of each prediction task from the large language model is fairly above random guessing, especially in a zero-shot setting with no task-specific finetuning, but are realistically too low to be useful in practice. There is much room for improvement, but also a great potential to be generalized to more tasks, multi-user or multi-tasking environments.

Third, there is a big difference between hands-on task guidance versus watching a "How-To" YouTube video. If we look closely, most of the model generated responses are instructions, instead of more situation-aware and personalized responses such as question answering and confirmations. It is easy to repeat recipe instructions or perhaps offer a little more detail leveraging the knowledge base, but harder to communicate with the user in a more situation-relevant way. Sometimes, a simple confirmation can boost the user's confidence, and worth a lot more than repeating what the user should do.

Furthermore, besides user intent and hesitation, there are other types of user states that can be helpful but are not currently modeled by our system. Are they familiar with the recipe? Do they look confused? Are they getting annoyed by all the instructions? Are they emotionally stable? All of these can change the way the instructor should talk. It is hard to categorize and collect users' mental states at each time point throughout the task. Sometimes experienced human instructors can infer through user actions and utterances, but the large language models have not demonstrated a strong ability in that way in our experiments.

Lastly, on the model architecture side, we leveraged some of the best performing multimodal and language models currently available, translated the rich audiovisual context collected in WTaG to language, and used LLM as the only reasoning engine to generate guidance. Language is too concise of a medium to offer context, and sometimes a picture is worth a thousand words. The current multimodal foundation models have limited reasoning capabilities to answer complicated questions and offer accurate timely guidance. This opens a wide variety of future opportunities for improving multimodal foundation models for situated task guidance.

## Limitations

As exciting as this work is, there are several limitations we would like to acknowledge and hope to improve for future works.

One of the biggest challenges we faced was evaluation. There isn't a great method to comprehensively evaluate generative language models in a scalable way. There are hardly many quantitative ways to capture the syntax, semantics, informativeness, relevancy, etc of the outputs, and human evaluation is costly and subjective. In this work, we tried to include both aspects, by evaluating models' performance on several classification tasks, as well as human evaluation on the guidance context. At the end of the day, we just need a functioning robot that communicate with human, and guide us step by step to make a cake. But to get there, we believe these are some of the milestones that the models should be able to achieve.

Another challenge related to evaluation was how to evaluate an interactive system in a real world setting. Every perception understanding can change models' decision making, and every decision prediction could completely change what would happen next. In our work, we had to focus on single step decision prediction given the ground truth interaction history. Our recorded dataset serves as resource to train/fine-tune/in-context learn/evaluate models on these decision points. Once a decent performing model is ready, we would love to conduct real-time human-bot interaction evaluations.

The third challenge is scale and resources. Our dataset contains 10 hours of recordings and 3 recipe tasks. A more robust baseline would ideally include a more diverse pool of users and instructors, more tasks, different environment setups, and etc. This would take collected effort and we are planning on expending our work in the future.

On the computation side, due to limited resources, we only ran the full experiments on the latest version of the GPT3.5-turbo-0301 model instead of GPT4. Querying large language models can be costly especially when we are querying at a high frequency (every few seconds per video) with a growing sized prompt. Meanwhile, there will always be another model version update in a few months that may achieve higher performance. However, we believe a lot of our observations still stand as the visual to language translation is a tough bottleneck to offer appropriate context, hallucination is still a big challenge for generative models, and

situated personalized guidance is yet to be ideal.

About the experiment results, there is a fair amount of randomness depending on what prompt was used, which version of the model it is, how we set all the hyperparameters for each model, etc. We reduced the temperature to be zero to minimize the randomness, and yet the exact same prompt can lead to different responses due to the inherent randomness of the LLM. An exhaustive prompt tuning and hyperparameter search is almost impossible and most of them can hardly be measured quantitatively. We reported several prompt and hyperparameter tuning results in Section 6, and ran the same prompts through ChatGPT three times. We would try to conduct larger scale tuning and evaluations in future works when resources are available.

Last but not the least, we experimented two vision processing methods, one for frame image captioning, and the other for object segmentation and object state detection. We chose these models as they've been reported to demonstrate state of the art performance recently in related tasks, also with a minor consideration of the processing speed, since the task in nature is ideally real-time. Nevertheless, there are a lot more models out there that can extract more fine-grained information, such as hand gestures, object segmentation, object tracking, or can take advantage of temporal video information instead of still frame inputs. We are planning on exploring more vision or other modality processing models for our task.

## Ethics Statement

The Institutional Review Board (IRB) of our institution approved this human subjects research before the start of the study. The WTaG dataset contains identifiable data (audio), but no facial identifiable data as all videos are first person views of the task environment. Members of the study team served as the human instructor, and recruited human subjects as the users. The human subjects were prepared the experiment setup, how to use the augmented reality headset, and potential risks before the experiment, and were debriefed after the recording. The consent to video and audio data for publication is optional and we will share the consented de-identified subset of the data (video and dialog ASR) when the paper is published. All data are stored in password protected internal servers with restricted access to only the study team. The hu-

man evaluations were conducted by 3 members of our study team, including two genders and two races. All collected data were filtered for content moderation during the ASR annotation stage. The generated utterances abide by the OpenAI content policy.

This work intends to have a positive impact on the society as the goal is to design a system that can assist human to complete tasks in a more personalized way. The impact of this work on the final goal is limited as we generated pre-recorded videos to study the baselines of the challenge, but there is the potential for the future models to be misused, or offer ill-intended guidance to human. We hope future works take responsible AI into considerations while designing human centered interactive systems.

## Acknowledgements

This work is supported by DARPA PTG program HR00112220003. We would like to thank the anonymous reviewers for their valuable comments and suggestions.

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

## A Appendix

### A.1 Sample Prompt

You are an instructor guiding a user to complete the task of Pour-over Coffee

[Recipe Content]

Chat History:
Time: 34.6, User: I wanna make sure I'm puring ten ounces
Time: 36.4, You: it should be twelve ounces of water

It has been 38.2 seconds into the recipe
Scene description: a person is preparing food on a table
OR
The user is interacting with Measuring cup
The Measuring cup is filled with water
User said oh twelve ounces ok

Answer the following questions:
1. What is their dialog intention? Choose among Question, Answer, Confirmation, Hesitation, Self Description, and Other
2. Which step is the user at? For example Step 3
3. Should you say anything? Yes or no
3.1. If yes, what would you say?
3.2. If yes, choose your dialog intention among Instruction, Confirmation, Question, Answer, or Other
3.3. If your dialog intention is Instruction, is it about current step, next step, details, or mistake correction
4. Did the user make a mistake? Yes or No? If yes, choose among wrong object, wrong state, wrong action
Answer:

Recipe: Pinwheels

Ingredients
1 8-inch flour tortilla
Jar of nut butter or allergy-friendly alternative (such as sunbutter, soy butter, or seed butter) Jar of jelly, jam, or fruit preserves

Tools and Utensils
cutting board
butter knife
paper towel
toothpicks
12-inch strand of dental floss plate

Steps
1. Place tortilla on cutting board.
2. Use a butter knife to scoop nut butter from the jar. Spread nut butter onto tortilla, leaving 1/2-inch uncovered at the edges.
3. Clean the knife by wiping with a paper towel.
4. Use the knife to scoop jelly from the jar. Spread jelly over the nut butter.
5. Clean the knife by wiping with a paper towel.
6. Roll the tortilla from one end to the other into a log shape, about 1.5 inches thick. Roll it tight enough to prevent gaps, but not so tight that the filling leaks.
7. Secure the rolled tortilla by inserting 5 toothpicks about 1 inch apart.
8. Trim the ends of the tortilla roll with the butter knife, leaving 1⁄2 inch margin between the last toothpick and the end of the roll. Discard ends.
9. Slide floss under the tortilla, perpendicular to the length of the roll. Place the floss halfway between two toothpicks.
10. Cross the two ends of the floss over the top of the tortilla roll. Holding one end of the floss in each hand, pull the floss ends in opposite directions to slice.
11. Continue slicing with floss to create 5 pinwheels.
12. Place the pinwheels on a plate.

---

Recipe (Simplified): Pinwheels

Take out a tortilla
Scoop choice of spread. Spread the spread.
Clean the knife.
Scoop a different spread. Spread.
Clean the knife.
Roll the tortilla.
Insert toothpicks.
Trim the ends of tortilla
Cut tortilla roll
Transfer pinwheels.

Figure 9: Sample Recipe and Its Simplified Version

| Frames | User | GT Inst | Lan | Sce | Obj |
|---|---|---|---|---|---|
| | Ok I don't drink coffee so I've never prepared coffee actually... | | Remind them of the next step which is step 2. | | Sure, no worries at all! |
| | Uh, filter cone. What is this? | Yeah the filter cone is the metal thing. | This is filter cone. It's part of the dripper that holds the paper filter. It's okay if you haven't seen this before. We'll move at a pace comfortable with you. | This is a paper filter. You need to fold it according to step 3. | Explain that the instrument is the kettle. |
| | Fold it twice? | Yeah. | Yes you fold the paper filter twice to create a cone. | It looks like you are taking out the coffee beans again. | Please unfold the filter again, and fold it in half to create a semi-circle, then fold in half again to make a quarter circle. Place it in the filter cone. |
| | Like this? | Yeah. You fold it again. | | Yeah the coffee ground should be the consistency of coarse sand. | Yes. That paper filter goes in the metal cone. After folding it in half, you need to fold it again so it'll shape like a cone. |
| | In half okay | | You should carefully follow the steps mentioned. | Fold it in half to produce semi-circle and in half again to create a quarter circle then put the filter into the dripper. | Fold the filter in half first and then again. |
| | Uh... | Yeah and now spread. | You should spread out the filter so it forms the shape of the cone to let the coffee drip through into the mug. | | Place the paper filter in the filter cone and spread it open to create a cone shape. |

Figure 10: Sample Data from WTaG: Coffee

| Frames | User | GT Inst | Lan | Sce | Obj |
|---|---|---|---|---|---|
| | | Oh you should place the tortilla on a cutting board. | Spread the nut butter on to the tortilla, leaving ½ inch uncovered at the edges. | | Spread the nut butter on to the tortilla, leaving ½ inch uncovered at the edges. |
| | Oh okay. | | You should place the tortilla on the cutting board | Oh you should place the tortilla on the cutting board. | You should place the tortilla on the cutting board |
| | | Oh you should use a butter knife to scoop nut butter from the jar. | Place the tortilla on the cutting board. | Clean the knife by wiping with a paper towel. | You should place the tortilla on the cutting board |
| | Oh oh yes. | | Oh you should use a butter knife to scoop nut better from the jar. | Use a butter knife to scoop nut butter from the jar. Spread nut butter onto tortilla, leaving ½ inch uncovered at the edges. | Secure the tortilla by inserting 5 toothpicks about 1 inch apart. |
| | | It's on the right with the red lid. | Clean the knife by wiping with a paper towel. | Use a butter knife to scoop nut butter from the jar. Spread nut butter onto tortilla, leaving ½ inch uncovered at the edges. | Please clean the knife by wiping with a paper towel |
| | Oh oh yes. | | Do you have a cutting board? | You are doing great! Just remember to use a butter knife to scoop the nut butter from the jar. | You should tell the user where the nut butter is. |

Figure 11: Sample Data from WTaG: Pinwheel