# OpenReview forum: "Can Foundation Models Watch, Talk and Guide You Step by Step to Make a Cake?"
_EMNLP/2023/Conference — EMNLP 2023 Findings_

### Official Review · Reviewer_nrzY · 2023-08-04

**Soundness:** 3

**Excitement:**

3: Ambivalent: It has merits (e.g., it reports state-of-the-art results, the idea is nice), but there are key weaknesses (e.g., it describes incremental work), and it can significantly benefit from another round of revision. However, I won't object to accepting it if my co-reviewers champion it.

**Missing References:**

Not a necessity but could have also cited: "Situated and Interactive Multimodal Conversations" – situated multimodal interactions in the fashion domain.

**Paper Topic And Main Contributions:**

In this paper is proposed a new dataset for multimodal task guidance called Watch, Talk and Guide (WtaG), as well as some baselines based on foundational models.

This dataset is composed of about 10 hours of egocentric videos of users performing 1 recipe out of 3 considered, while guided by a human instructor in a live setting.
According to the authors, one of the things that differentiates this dataset from previous work is the interactivity and annotation of both dialog acts and mistakes.
The authors define two main tasks: 1) user environment understanding, and 2) instructor decision-making - casting most of these tasks into a classification task for easier evaluation.

Regarding the methods used, the authors considered a foundational model (GPT-3.5) without task-specific fine-tuning. Given that the model only works with text, the authors have two additional input strategies. The first is based on scene description using BLIP-2, and the second is based on object and state detection using fine-grained scene information.

The results showed low accuracy attributed to the difficulty of the task, and it being a zero-shot setting. The results also showed that visual information was mostly unhelpful to the model, given that they were too generic or irrelevant, generally resulting in hallucinations.
An additional study also showed that the quality of task guidance is subjective, hinting at the possibility of future work in modeling the user as well.


**Questions For The Authors:**

A) Who are these subjects, how were they recruited, what are their demographics, and experience in the tasks (cooking).

B) What is the language of the dataset? I assume it is English but it should be clarified.

C) Have you tried training a model on this data instead of using a zero-shot setting?

**Reasons To Accept:**

1. The dataset presented is novel and targets the interesting scenario of task guidance in a real setting.
2. Annotation of both sides of the dialogue and classification of the main types of interactions and mistakes.

**Reasons To Reject:**

 1. The approach presented relies on the foundational model’s ability to decipher and understand the input many times with insufficient context. It would have been good to have, as a baseline, a model trained on this data, given that all of the tasks are annotated.
2. The paper does not mention the origin and experience of the subjects utilized in the data collection experiment.
3. I would have liked to see more examples of the datasets' conversations and a more complete analysis of the comprehensiveness and interactivity between user and instructor.

**Reproducibility:**

3: Could reproduce the results with some difficulty. The settings of parameters are underspecified or subjectively determined; the training/evaluation data are not widely available.

**Reviewer Confidence:**

4: Quite sure. I tried to check the important points carefully. It's unlikely, though conceivable, that I missed something that should affect my ratings.

**Typos Grammar Style And Presentation Improvements:**

Typos and Grammar:

Line 3 - assist humans in various

Line 14 – we leverage a large language model backbone

Line 37 – to guide the human through

Line 66 – quantitatively

Line 92 – review the last sentence

Line 126 – videos

Line 150 – review the sentence and “open-world”

Line 187 – Figure 2 and in the Appendix (specify where in the appendix)

Line 937 – “medium” – is this median or average

Line 99 – ranges from

Line 451 - To to

---

> ### Author Rebuttal · Authors · 2023-08-27
>
> Thank you very much for recognizing the novelty and annotation efforts of this work! We would also like to thank the reviewer for all the detailed reviews and helpful suggestions! Here are a few things that we can address here and we definitely welcome more questions during the discussion period if any questions remain.
>
> **[Train on This Data instead of Zero-Shot, Q-C]:**
>
> It definitely came across our mind to see if we can train or fine-tune a model to achieve better performance. The main consideration that led to the decision was that we are aiming for the **generalizability** of the system which hopefully can be extended to any arbitrary given task manual or recipe, a large scope for which it is impossible to collect comprehensive training data.
>
> With this goal in mind, a model would either have to learn a set of predefined tasks, or in our method, leverage the world knowledge of pre-trained large models (GPT, CLIP, etc) and in-context learning capabilities.
>
> Training or fine-tuning on this dataset, on the other hand, would entail limiting the vocabulary size of object detections, tailoring the reasoning model towards a specific set of tasks, and overfitting the language generation model towards the set of human user preferences in our experiments. *It might improve the performance on this dataset, but might not be the answer to the ultimate goal that we strive for.*
>
> That being said, training definitely has its merits and as observed, the current system has a lot to improve on. We hope this dataset can help with model improvements in various ways in future works.
>
> We will clarify these in the revision.
>
> **[Human Subject Demographic, Q-A]:**
>
> Our IRB approval limits the types of subject data we can collect and share, but all subjects were over the age of 18, English-speaking college students from different cultural background with normal or corrected-to-normal vision, recruited through messaging platforms.
>
> We did not explicitly survey their experience in cooking as one single self-identified value might not be as informative as we would hope for. Instead, all the subjects were given a simplified recipe which may not be something they are familiar with. For example, the cultural background of the subjects varies and an experienced cook for one cuisine doesn’t necessarily entail they are familiar with recipes from other regions; or the subject’s familiarity with individual recipe steps could be very different from their familiarity with the rest of the recipe and a regular coffee drinker might stumble into (how to use) a new food scale.
>
> We will clarify this and offer as many details about the users as we could in the revision.
>
> **[More Examples and Analysis]:** We are happy to include more examples in the revision. The dataset and code will also be available upon publication. We were hoping Figure 2b offers some insights on the interactivity between users and instructors, but will reorganize it in a more intuitive way and include more comprehensive analysis.
>
> **[Q-B: Dataset Language]:** Sorry about that - all data collected is English.
>
> We will fix all the typos listed, double check the rest, and include the additional references in the revision. Thanks a lot!

---

### Official Review · Reviewer_gqEJ · 2023-08-05

**Soundness:** 3

**Excitement:**

2: Mediocre: This paper makes marginal contributions (vs non-contemporaneous work), so I would rather not see it in the conference.

**Paper Topic And Main Contributions:**

The paper explores utilizing foundational models for task guidance in a cake making scenario. The work contributes to the literature by developing a dataset consisting of conversational interactions between a guide and a task performer. The task pipeline consists of vision modules tasked with understanding the visual content and generating context for the LLMs which is later converted to a prompt to allow generation of utterances to be used for task guidance. The authors perform evaluations on the perceptions and the model performance for the pipeline and infer that, while the LLM show promise there is much left to be done on the multimodal front to perform task guidance (for cake making). There are also differences that exist in real-time task guidance when compared to performing more deterministic activity recognition.

**Questions For The Authors:**

- Can the narrations/generations be further used to improve the models rather than a zero shot inferencing approach? This could be a more interesting approach highlighting the need to understanding the domain.
- LLMs are quite good at recommending recipes and guidance related to recipes.
- How is the the time synchronization issues handled especially with lags in the generations and the time sensitiveness?

**Reasons To Accept:**

- Task guidance is a challenging problem. The problem has multitude of complexities not limited to the capabilities of the model but also on the user perception (annoyance for instance). The work very promisingly identifies these limitations and provides a valuable dataset.


**Reasons To Reject:**

- Task guidance in a situated setting has remained a challenge. (See for example: Conversational image editing: Incremental intent identification in a new dialogue task; Manuvinakurike et al., 2018;CAISE: Conversational Agent for Image Search and Editing; Kim et al. 2022). One of interesting aspect of real-time task guidance is that of real-time or incremental nature of the problem. Some steps in the guidance last a few seconds while a few last minutes. Contextual variables (type of dough or ingredients for instance), personal preferences etc. all play a role. The work doesn't highlight the nuances involved in these domains which could be more valuable for the community and potentially allow development of novel applications.
- "When to ask" is a problem involving turn-taking elements which involve a lot of contextual understanding. The literature shows this to be a challenging problem and is rather a more involved process.


**Reproducibility:**

3: Could reproduce the results with some difficulty. The settings of parameters are underspecified or subjectively determined; the training/evaluation data are not widely available.

**Reviewer Confidence:**

5: Positive that my evaluation is correct. I read the paper very carefully and I am very familiar with related work.

---

> ### Author Rebuttal · Authors · 2023-08-27
>
> Thank you very much for recognizing the problem that this work is tackling is challenging and that this work promisingly identifies several limitations of the current methods and provides a valuable dataset. We also appreciate all the insightful suggestions and comments which we address below.
>
> First of all, we would like to clarify that our dataset contains three tasks instead of just cake making: pinwheels, pour-over coffee, and a microwaved mug cake. For example, A.4 is a sample prompt for coffee making, and Figure 12 is a recipe for pinwheels. We ultimately aim to design a task guidance system that has the generalizability to be extended to any given task manual/recipes (Section 2.1), more than just making cakes or within the cooking domain. We will clarify this in the revision.
>
> **[Challenging]:** We fully agree with the reviewer that situated task guidance is very challenging. There were a lot of angles to consider, and it was indeed difficult to find a niche point to start. That was exactly why we decided to collect this dataset with both human instructors and users as the ‘gold standard’ to study and understand the problem better. We want to mimic the task in the most realistic setting (natural language conversations, natural mistakes, synchronized multimodal inputs, an arbitrary task manual given to the LLM etc), which we haven’t found similar resources in the previous works, and we documented all the data along the way. For baseline method comparisons, we focused on one of the technical difficulties (multimodal context incorporation into LLMs) that we observed to experiment on, when there are indeed a lot of other nuances that deserve attention. While this effort lays a foundation, we hope that this work and dataset can enable future work to study the depth and breadth of the problem with all of its numerous complexities.
>
> **[Q1: Further Improvement with This Data instead of Zero-Shot]:** Absolutely! It’s possible that more tailored models are needed for complex tasks like this. For example, as observed in Section 6.2 human evaluations, task guidance can be a fairly personalized experience. Guidance generation can be further trained to accommodate situation specific, user specific adjustments. The dataset we collected allows us and others to further the studies on interesting directions like this in future works.
>
> **[Q2: LLMs are good]:** Yes they are. One big challenge that we observed with LLMs through our experiments is to offer situated and personalized guidance with multimodal context of what is happening, how is the user doing, etc. Our final goal is to design a fully interactive system that can accommodate this, and we hope this work serves as a starting point on what the current existing methods can achieve so far and to identify several key areas for future works to improve on.
>
> **[Q3: Time Synchronization]:** That is indeed a very important issue that we have been keeping in mind while designing the system with the tradeoff of model performance and model processing speed. About this work specifically, these issues are less pertinent as we only evaluate them based on collected data, with the observation that the LLM based methods still have a large room for improvement even without considering these efficiency challenges. However, these issues will indeed become significantly more important in the real-time interaction settings. As such, we hope this work and the dataset that we collected can help future model design, performance and efficiency comparisons.
>
> We will clarify all the issues mentioned above and incorporate the references in the revision. Thanks a lot for all the suggestions!

---

### Official Review · Reviewer_i9MT · 2023-08-09

**Typos Grammar Style And Presentation Improvements:** Some typos
**Soundness:** 3

**Excitement:**

4: Strong: This paper deepens the understanding of some phenomenon or lowers the barriers to an existing research direction.

**Paper Topic And Main Contributions:**

This paper investigates a very challenging task of building a zero-shot guidance model by leveraging LLMs. The authors propose a new dataset WTaG which mimics the real-world situations to testify whether the LLMs can fulfill the above task. The task is very difficult, especially involving multimodal information, i.e., textual and visual data. And the experiments can show the promising performance of LLMs although there is still much room for improvement.

**Reasons To Accept:**

1. The task of building a zero-shot multimodal task guidance system by leveraging LLMs is quite interesting and bold.

2. The authors propose a new dataset which is promising to facilitate this kind of research line.

3. The experiment results can inspire more related researches.

**Reasons To Reject:**

1. Unlike LLMs, in my opinion, multimodal large models have not been shown that they have the ‘emergent abilities’.

2. The problem to resolve is not clearly defined. And the technical contribution is limited. As a benchmark experiments on a new dataset, comparison with baselines is missing, e.g., using various large models.

**Reproducibility:**

3: Could reproduce the results with some difficulty. The settings of parameters are underspecified or subjectively determined; the training/evaluation data are not widely available.

**Reviewer Confidence:**

2: Willing to defend my evaluation, but it is fairly likely that I missed some details, didn't understand some central points, or can't be sure about the novelty of the work.

---

> ### Author Rebuttal · Authors · 2023-08-27
>
> We thank the reviewer for recognizing this is a "bold" research, and that our dataset will be promising to inspire future work. We also appreciate the valuable feedback and questions which we will address below. We welcome more questions during the discussion period if any of our response is not clear.
>
> **[Problem Definition]:** At the high level, the goal is to design an interactive system that can offer timely personalized and situated guidance by watching and communicating with the user. In Section 4, we define the problem and key characteristics of it that we can evaluate models for.
>
> More specifically, given inputs: 1) user's first person view video, and 2) user's utterances, both up until time point $t$, predict the followings:
>
> [User and Environment Understanding]
> 1) User's utterance intention (if any), choosing from a set of choices
> 2) Current step, choosing from recipe steps
> 3) Is there a mistake at time $t$, yes or no. If yes, then predict the mistake type, choosing from a set of choices
>
> [Instructor Decision Making]
> 1) Should the instructor talk at time $t$, yes or no
> 2) If yes to 1), Instructor's utterance intention, choosing from a set of choices
> 3) If yes to 1) and 2) type is "Instruction", Instructor's instruction type, choosing from a set of choices
> 4) If yes to 1), the instructor's utterance
>
> We will add mathematical definitions of the tasks and clarify this part in the revision.
>
> **[Technical Contribution]:** As the reviewers mentioned, the problem has numerous aspects to consider and can be challenging to find a starting point. Our main goal with this work is to collect a dataset with both human users and instructors as the ‘gold standard’ to study and understand the problem better. We hope this work could serve as a starting point in the most realistic setting (natural language conversations, natural mistakes, first person view, synchronized multimodal inputs, etc) for future work to improve upon.
>
> **[Baseline Comparisons]:** For baseline comparisons, we aim to construct a complete pipeline that takes multimodal inputs, and outputs language guidance given any arbitrary task. As observed through the experiments, a bigger challenge of leveraging LLMs as backbones is incorporating visual context, rather than generating guidance. We therefore focused on experimenting with three different methods of incorporating multimodal inputs for the LLM backbones. There are, however, indeed a lot of other angles that can be further investigated (e.g. different language models, different object and state detection methods, different multimodal fusion methods, contextual variable controls, user’s preferences, etc) and we plan on further studies in future works.
>
> **[Emergent Abilities]:** If this was in reference to Section 2.2 (Ln 104), we believe we were referring to the emergent abilities of LLMs there instead of multimodal large models, but we will rephrase this section to be clearer.
>
> We will fix the typos and double check the rest in the revision. Thanks for pointing them out!

---

### Meta-Review · Area_Chair_t4Uk · 2023-09-10

**Recommendation:** 3

**Metareview:**

In general, reviewers agreed that this paper provided sufficient support for its major claims, although some minor points could benefit from further support or clarification.  They exhibited mixed enthusiasm for seeing the paper at the conference, with excitement ratings ranging from mediocre to strong.  The authors provided additional support for many of the claims that reviewers found to be more weakly justified, and they promised to update the revised manuscript with this additional information.

**Summary of Reviewer Feedback and Discussion:**
- **Reviewer i9MT** found the proposed task to be interesting and bold, and they appreciated that the authors proposed a dataset to facilitate research towards that task.  They also felt that the experimental results may inspire additional related research.  However, they were unconvinced that multimodal models have demonstrated evidence of emergent abilities, and they felt that the paper's problem definition was unclear.  They also felt that the paper's technical contributions, including comparisons of different potential baselines for the new dataset, were limited.  The authors clarified their problem definition in the rebuttal and promised to formalize it more clearly in the revised manuscript.  They also clarified that their technical contribution is the new dataset, and agreed that there are many more model comparisons that they can perform in the future to study their problem from different angles.  As a point of clarification, the authors noted that they intended to refer to emergent abilities of LLMs, rather than large multimodal models.
- **Reviewer gqEJ** thought that the proposed task was challenging and that the provided dataset will be useful for investigating it further, but they felt that the paper didn't adequately highlight the nuances involved in associated problem domains.  They also felt that the authors may have missed some relevant literature, and they raised several clarifying questions and a clarifying comment.  In their rebuttal, the authors clarified the variety of tasks in their dataset and noted that they chose one specific technical challenge as the focus of their experiments but that they agree that there are many other nuances to explore as well.  They also responded to Reviewer gqEJ's other clarifying questions, and promised to clarify these points in the revised manuscript.
- **Reviewer nrzY** thought that the proposed dataset was novel, high-quality, and targeted an interesting scenario.  However, they wished that there had been a baseline trained on this data rather than only baselines using foundation models.  They also requested more details pertaining to both the dataset itself and the subjects who participated in its development.  In their rebuttal, the authors clarified that they hadn't used a baseline trained on their dataset because they were interested in generalizability beyond their dataset alone.  They also responded to Reviewer nrzY's other questions, provided additional details regarding their human subjects, and promised to include more examples and analyses in the revised manuscript.

---

### Decision · Program_Chairs · 2023-10-07

**Decision:**

Accept-Findings

**Comment:**

In general, reviewers agreed that this paper provided sufficient support for its major claims, although some minor points could benefit from further support or clarification.  They exhibited mixed enthusiasm for seeing the paper at the conference, with excitement ratings ranging from mediocre to strong.  The authors provided additional support for many of the claims that reviewers found to be more weakly justified, and they promised to update the revised manuscript with this additional information.

**Summary of Reviewer Feedback and Discussion:**
- **Reviewer i9MT** found the proposed task to be interesting and bold, and they appreciated that the authors proposed a dataset to facilitate research towards that task.  They also felt that the experimental results may inspire additional related research.  However, they were unconvinced that multimodal models have demonstrated evidence of emergent abilities, and they felt that the paper's problem definition was unclear.  They also felt that the paper's technical contributions, including comparisons of different potential baselines for the new dataset, were limited.  The authors clarified their problem definition in the rebuttal and promised to formalize it more clearly in the revised manuscript.  They also clarified that their technical contribution is the new dataset, and agreed that there are many more model comparisons that they can perform in the future to study their problem from different angles.  As a point of clarification, the authors noted that they intended to refer to emergent abilities of LLMs, rather than large multimodal models.
- **Reviewer gqEJ** thought that the proposed task was challenging and that the provided dataset will be useful for investigating it further, but they felt that the paper didn't adequately highlight the nuances involved in associated problem domains.  They also felt that the authors may have missed some relevant literature, and they raised several clarifying questions and a clarifying comment.  In their rebuttal, the authors clarified the variety of tasks in their dataset and noted that they chose one specific technical challenge as the focus of their experiments but that they agree that there are many other nuances to explore as well.  They also responded to Reviewer gqEJ's other clarifying questions, and promised to clarify these points in the revised manuscript.
- **Reviewer nrzY** thought that the proposed dataset was novel, high-quality, and targeted an interesting scenario.  However, they wished that there had been a baseline trained on this data rather than only baselines using foundation models.  They also requested more details pertaining to both the dataset itself and the subjects who participated in its development.  In their rebuttal, the authors clarified that they hadn't used a baseline trained on their dataset because they were interested in generalizability beyond their dataset alone.  They also responded to Reviewer nrzY's other questions, provided additional details regarding their human subjects, and promised to include more examples and analyses in the revised manuscript.